# Oogenesis in Women: From Molecular Regulatory Pathways and Maternal Age to Stem Cells

**DOI:** 10.3390/ijms24076837

**Published:** 2023-04-06

**Authors:** Kornelia Krajnik, Klaudia Mietkiewska, Agnieszka Skowronska, Pawel Kordowitzki, Mariusz T. Skowronski

**Affiliations:** 1Department of Basic and Preclinical Sciences, Faculty of Biological and Veterinary Sciences, Nicolaus Copernicus University, 87-100 Torun, Poland; 2Department of Human Physiology and Pathophysiology, School of Medicine, Collegium Medicum, University of Warmia and Mazury, 10-719 Olsztyn, Poland

**Keywords:** oocyte, oogenesis, embryogenesis, women, induced pluripotent stem cells

## Abstract

It is a well-known fact that the reproductive organs in women, especially oocytes, are exposed to numerous regulatory pathways and environmental stimuli. The maternal age is one cornerstone that influences the process of oocyte fertilization. More precisely, the longer a given oocyte is in the waiting-line to be ovulated from menarche to menopause, the longer the duration from oogenesis to fertilization, and therefore, the lower the chances of success to form a viable embryo. The age of menarche in girls ranges from 10 to 16 years, and the age of menopause in women ranges from approximately 45 to 55 years. Researchers are paying attention to the regulatory pathways that are impacting the oocyte at the very beginning during oogenesis in fetal life to discover genes and proteins that could be crucial for the oocyte’s lifespan. Due to the general trend in industrialized countries in the last three decades, women are giving birth to their first child in their thirties. Therefore, maternal age has become an important factor impacting oocytes developmental competence, since the higher a woman’s age, the higher the chances of miscarriage due to several causes, such as aneuploidy. Meiotic failures during oogenesis, such as, for instance, chromosome segregation failures or chromosomal non-disjunction, are influencing the latter-mentioned aging-related phenomenon too. These errors early in life of women can lead to sub- or infertility. It cannot be neglected that oogenesis is a precisely orchestrated process, during which the oogonia and primary oocytes are formed, and RNA synthesis takes place. These RNAs are crucial for oocyte growth and maturation. In this review, we intend to describe the relevance of regulatory pathways during the oogenesis in women. Furthermore, we focus on molecular pathways of oocyte developmental competence with regard to maternal effects during embryogenesis. On the background of transcriptional mechanisms that enable the transition from a silenced oocyte to a transcriptionally active embryo, we will briefly discuss the potential of induced pluripotent stem cells.

## 1. Introduction

Sexual reproduction is a process during which many organisms ensure the continuity of the species by producing offspring [1]. It arises in connection with the fusion of haploid gametes from parental organisms, and as a consequence, a zygote is formed, from which an embryo develops [2,3]. The process of gamete formation to create a progeny organism is a highly complex process regulated by many intra- and extraovarian factors in the female embryo. Previous studies have provided knowledge with regard to aging and infertility. Thanks to discoveries from the past decades, nowadays, we know more about the molecular events taking place during the process of oogenesis and embryogenesis. Herein, we aim to gather knowledge regarding the transcriptional regulation of late oogenesis and early embryogenesis. We intend to point out the regulation of transcripts during oogenesis since it appears to be crucial for oocyte maturation and fertilization later in life in women [4]. Further, we aim to describe the relevance of regulatory pathways during oogenesis in women and focus on the molecular pathways of oocyte developmental competence with regard to maternal effects during embryogenesis. On the background of transcriptional mechanisms that enable the transition from a silenced oocyte to a transcriptionally active embryo, we will briefly discuss the potential of induced pluripotent stem cells at the end of this review.

## 2. Oogenesis, Oocyte Growth, and Oocyte Maturation

Oogenesis is the process during which the formation of the body’s largest cell, the oocyte, takes place [5,6]. This multi-step process consists of many interactions between the developing oocyte and the granulosa cells and cumulus cells surrounding the oocyte [7]. Oogenesis begins in the fetal ovaries when oogonia are developed from primordial germ cells (PGC), as soon as the development of the embryo progresses, in approximately the 12th week of gestation in women [5,8]. They then undergo well-organized processes, such as the pairing of homologous chromosomes and the crossing-over process of chromosomes [9]. Genes that are important for chromosome segregation include *STAG3*, which encodes protein stromal antigen 3, or BUB1B, which encodes Mitotic Checkpoint Serine/Threonine Kinase B [10]. During the prophase of meiosis I, homologous chromosomes undergo recombination, during which DNA double-strand breaks (DSB) occur [11]. This is when the proliferation phase of the oogonia takes place. Following, these cells arrest at the end of prophase I, and remain dormant in this state until the periovulatory phase [1,12]. It has been reported that the number of oocytes in newborn girls is about 1 million, and at the timepoint of reaching puberty, this number has already decreased to 400,000 [6].

To determine the reproductive potential of the ovaries, the so-called ovarian reserve is commonly used in human fertility clinics, reflecting the number of remaining follicles bearing the oocytes [12]. The reproductive machinery prepared in this way remains transcriptionally inactive from the start of follicle growth until fertilization; thereafter, activation of the zygotic genome takes place (ZGA) [12,13]. In some women, a sudden decrease of the ovarian follicles after puberty can occur, diminishing the ovarian reserve and symptoms of irregular cycles or an early onset of menopause might be present [6]. The process of oogenesis is regulated by the neurotrophin signaling pathway. Polypeptide growth factors, called neurotrophins (NT), and their receptors in cell membranes have been shown to control processes such as follicle formation and growth [14]. Oogenesis is also under the control of endocrine, paracrine, and autocrine factors. One of the factors influencing the process of oogenesis is adipokines, i.e., substances secreted by adipose tissue [15].

Leptin is one of the adipokines that is present both in mature follicles and in the earlier stages of follicle development, and its presence supports, among others, the development to the stage of primary follicles [16]. It has been confirmed that treatment with leptin administered in low doses provides a chance to accelerate the growth and maturation of ovarian follicles [17]. In addition, previous research has shown that mice showing a deficiency of leptin had a smaller number of follicles [18]. However, it has also been reported that an extrinsic apoptosis pathway can be induced via the upregulation of caspase 3 (CASP3) as a result of acute leptin treatment [19]. Another essential adipokine affecting the process of oogenesis is adiponectin. The deficiency of the latter-mentioned one reduces the number of ovulated oocytes [20]. Different expressions of resistin can be observed in human ovarian follicles at different stages of development, which provides evidence for the participation of this adipokine in the process of oogenesis [21].

There is no doubt that oocyte development is a very energy-intensive process. The energy provided by the oocyte is especially important in the first stages of embryo development [22]. The main sources of energy are fatty acids and glucose. However, research has shown that fatty acids are the primary source of energy for oocytes, and not glucose. Oocytes secrete various paracrine factors, such as bone morphogenic protein 15 (BMP15) and growth differentiation factor 9 (GDF9) [23]. Paracrine factors secreted by fully developed oocytes have the ability to upregulate the process of glycolysis. It has been reported that the removal of oocytes from the COCs (cumulus-oocyte complexes) led to a decrease in the level of transcripts encoding for the enzymes Pfkp—phosphofructokinase and Ldha—Lactate dehydrogenase A, which are both glycolytic enzymes [24].

Transcriptional activity and gene expression regulation are topics of particular interest for reproductive biologists and reproductive medicine specialists [2,25]. Many factors affect transcriptional activity, including DNA methylation, histone modification, or changes in the histone composition of the nucleosome [26]. DNA methylation, which is a mechanism of epigenetic regulation, is observed during the transition of follicles to the antral stage, at the end of the growth phase, which corresponds to the time of their transcriptional silence [27,28].

There are several studies on proteins that have been shown to be indicators for oocyte growth. In cattle, a protein called connexin 43 (Cx43) has been recognized as a marker of oocyte developmental competence, and connexin 45 (Cx45) and connexin 60 (Cx60) are the main connexins expressed during folliculogenesis in pigs [29]. Apelin plays a very important role in the growth of the ovarian follicle, affects angiogenesis, and the formation of the corpus luteum, and may affect the proliferation of granulosa cells [30]. At the start of folliculogenesis, there is an increase in the expression of the gene encoding tyrosine kinase receptor—KiT in the oocyte [23]. A downstream effector of KiT is phosphatidylinositol 3-kinase (Pi3K), which, once it is activated, phosphorylates serine/threonine kinases. The latter-mentioned enzymes are involved in oocyte survival and proliferation [31]. As previously mentioned, BMP15 and GDF9 expressed in oocytes promote follicle differentiation [32].

It has been reported that an increase in the H3K4me3 modification, which is considered as a marker of active transcription, indicates histone methylation during oocyte growth. Recent studies indicate that it reaches its maximum level at the time point of silencing transcriptional activity, i.e., at the end of the growth phase [33,34]. Down-regulation of promoters of polymerase II (responsible, among others, for the production of pre-mRNA) is also indicated to be a result of global transcription silencing at the end of oocyte growth. The end of this phase will be dominated by processes of transcriptional silencing and degradation of some mRNAs [4]. Notably, oocyte maturation is a consequence of the interaction between the oocyte and the granulosa cells [23]. Most of the mechanisms discussed in this review are more of a sequence of events than a single event. In vitro maturation of human oocytes is one of the most difficult hurdles when performing an IVF procedure [5,35]. The time at which the egg reaches nuclear maturity and acquires meiotic competence occurs at the same time as the antrum is formed, which is specific to mammal species [36]. This occurs when the oocyte has grown to 80% of its final size [4]. It has been estimated that mouse oocytes that have reached meiotic competence contain approximately 200 times more RNA than a somatic cell, of which 10–15% will be pre-mRNA. What happens to this mRNA in the oocyte depends mainly on regulatory proteins and ribosomes, while after transcription, the poly-A tail will be added [4]. It is observed that polyadenylation is evolutionarily conserved in many species, i.e., insects (Drosophila), amphibians (Xenopus), fishes, and mammals [2]. The key proteins in this process are CPEB and CPEB1, the most present ones in the mouse and human oocytes, responsible for the control of polyadenylation and translation, as well as for the activation of translation. Furthermore, CDK1 (cyclin-dependent kinase 1) and Mos activate the MAPK cascade (mitogen-activated protein kinase) [33,37]. Interestingly, the maturation-promoting factor (MPF) is produced in oocytes with the association of Mos and CDK1 [12,26]. In mammals, meiotic resumption under the influence of LH (luteinizing hormone) and FSH (follicle stimulating hormone) meditates the activation of MAPK, which leads to the increased production of cAMP (cyclic adenosine monophosphate) [38,39].

During the entire oocyte maturation process, changes in the genome and expression are based on translation and degradation, rather than on transcription [37]. The occurrence of degradation of translated transcripts results in only half of the mRNA remaining after degradation until the oocyte reaches metaphase II, and only 30% will remain immediately before fertilization [12]. In mouse oocyte studies, downregulation of GPR3 has been shown to contribute to oocyte maturation [40]. Mural granulosa cells and granulosa-derived cumulus cells play a crucial role in oocyte development by supporting the cell with metabolites and regulatory signals [41,42].

## 3. Oocyte Release, Relevant Hormones, Environment, and Maternal to Zygotic Transition

Due to the lack of transcriptional activity, the oocyte relies heavily on synthesized proteins and RNAs of maternal origin to complete meiosis [13]. Despite differences between species, the processes of transcription, translation, and degradation of the maternal transcriptome are preserved functions. However, regardless of the species, the transcriptome is remodeled and then degraded since, as indicated above, this is necessary for proper embryo development [12]. A fully mature oocyte is characterized by elevated levels of cAMP, which is appears to be key to maintaining the oocyte in prophase I [4]. The completion of meiosis I occurs as a result of the pre-ovulatory release of LH, which results in chromatin condensation [43]. Then, asymmetric division occurs, during which the polar body is extruded, which is tantamount to the end of meiosis [4,13]. Before ovulation, the ovum will enter the metaphase of meiosis II, and will not complete it until fertilization [26]. The LH surge not only affects the completion of meiosis, the processes also initiated by the presence of this hormone lead to the transformation of the follicle, angiogenesis, or differentiation of granulosa cells (GC) and theca cells (TC) surrounding the oocyte. As the first responders, GC’s and TC’s produce mediators of inflammatory processes. Those mediators activate several cells that regulate proteolytic pathways. The consequence of all these events is the rupture of the follicle and the release of the egg [44]. At this stage, transcriptionally silent oocytes rely on maternally derived mRNA for differentiation, protein synthesis, fertilization, early development, and transcriptional reactivation [12].

Oocytes contain RNA and proteins, creating a unique landscape that is crucial during early development because that is the only thing it relies on at the first stages of fertilization, meiosis-to-mitosis transition, reprogramming, and the beginning of embryogenesis [34,45]. The maternal to zygotic transition (MZT) is a stage that a large number of investigated species go through, and is a stage that occurs at the beginning of early development [2,46]. After fertilization, the development of the embryo is not directly led by the newly formed zygote, and there is a gradual shift of control from the maternal genome to the zygotic genome [47]. The transition occurs as two activities, the task of which is to differentiate the fertilized oocyte into a totipotent embryo. The first is to dispose of all the maternal machinery that was necessary to prepare the oocyte for fertilization and at the beginning of embryogenesis, and the second is to allow ZGA to occur [47,48,49].

While discussing all the processes that take place in the egg cell to transform MZT, one should mention the complexes created by RNA and ribosome proteins binding them (RBP), proteins that regulate almost all of the RNA life [13,45]. They are accumulated in the oocyte during its development and their task is to regulate the maternal transcript, thanks to which it can remain stable for up to a year [33]. Previous studies indicate that individual RBPs are rather species-specific, but they have been described in detail for species such as C. elegans, D. melanogaster, D. rerio, Xenopus, and M. musculus. In addition to regulation and stabilization, RBPs are responsible for destabilizing maternal mRNA during MZT [33].

The oocyte that had prepared itself for the standstill period during the growth phase accumulated various proteins necessary to survive this state, as well as proteins responsible for repressing transcription. The genome is prepared for activation by the accumulation of activators. Two models have been proposed that attempt to explain the delay in transcription activation. One of these hypotheses suggests that there are no proteins required to activate transcription, while the other hypothesis postulates that such proteins are present, but so are the inhibitors that block the possibility of genome expression [50]. There are two other mechanisms of protein regulation proposed that are responsible for repressing and activating transcription [48,50]. The first is when a repressive protein deposited by the mother blocks transcription in the early embryonic stages after fertilization with subsequent cell division due to the increasing ratio of the amount of maternal mRNA to the volume of the cytoplasm (the amount of maternal material is constant and the cytoplasmic volume increases). The repressor is titrated until its concentration is below a threshold level under which transcription can be initiated. The second mechanism involves low concentrations of translation activator, which together with the progress of cell division and the process of polyadenylation and transcription of maternal mRNA, leads to the accumulation of activator protein, which, after reaching the threshold level, is able to initiate gene expression [48,50] (Figure 1). An example of a transcription factor that affects transcription regulation is the Tramtrack (TKK) transcription factor in D. melanogaster. Deficiency of this factor has been shown to result in premature transcription of the repressed gene, and it is deleted by its specific RBP, thus demonstrating the link between maternal mRNA removal and the onset of ZGA [33].

Another example of a factor that affects the change in the transcriptional state in the oocyte is OCT4 in humans, or its Zelda homolog (Zld), which are factors that activate zygotic expression genes. Studies have shown that in the case of D. melanogaster, the lack of the Zld factor causes the death of embryos even before the end of MZT [33,50]. For the translation process, access to the chromatin must be made possible and this is determined by the degree of its packing. After fertilization, the genome of mammalian species undergoes a process of global demethylation, which is a reversal of the process that was responsible for the transcriptional silencing of the genome at an earlier stage. Similarly, there is the appearance of H3K4me3, which is frequently located in transcription start sites (TSS) [33,50]. Therefore, it can be summarized that the genome is prepared for activation by the accumulation of activators, loss of repressors, and local changes in chromatin accessibility [33]. The re-activation of the transcription machinery will occur after fertilization of the oocyte at the zygotic genome activation (ZGA) stage [13]. It is not clear why ZGA does not occur immediately after fertilization but only a few hours later [12]. However, researchers indicate that this allows the sperm and egg genomes to merge [50]. ZGA is shown as a phase, not a single phenomenon, with two waves, a small and a large one. The small wave of ZGA begins when the first cleavage divisions occur, while the large wave occurs when many species show a break in the cycle of subsequent divisions [27,50]. As previously indicated, while the maternal transcriptome is crucial for proper fertilization and the start of the development of the embryo, at the stage of about 2 to 8 cells, its degradation is already completed depending on the species [46,51]. After ZGA is finished, the EGA (embryonic genome activation) starts [52].

Concentrations of steroid hormones, estradiol and progesterone, determine the potential fertility of women and affect many issues related to their health [53]. The dominant sex steroid hormones in women are estrogens, such as estrone (E1), estradiol (E2), and estriol (E3). Estradiol is the main circulating sex hormone before menopause, while estrone plays an important role after menopause [54]. The direct substrates for the synthesis of estrogens are testosterone and androstenedione [55]. Here, SNARE proteins (soluble NSF attachment receptor) are involved in the transport of cholesterol from lipid cells. Cholesterol is then converted to pregnenolone for further production of dehydroepiandrosterone (DHEA), which is an androgen precursor [56]. The site of estrogen synthesis in non-pregnant women is mainly the ovary, while in pregnant women it is the placenta [54]. Estrogens, among others, play a crucial role during the regulation of the menstrual cycle [57]. In addition, these hormones affect calcium and lipid metabolism and affect protein synthesis [58]. The level of circulating estradiol and anti-müllerian-hormone (AMH) is affected by the reproductive status of women. The AMH level especially decrease with advancing age and with an advancing decrease of remaining follicles, mirroring, therefore, the ovarian reserve (Figure 2) [54]. When discussing the hormonal regulation of the female reproductive cycle, the hypothalamic–pituitary–ovarian axis should also be mentioned. An important hormone of this axis is gonadotropin releasing hormone (GnRH), which is a hypothalamic hormone stimulating the anterior lobe of the pituitary gland to release follicle-stimulating hormone (FSH) and luteinizing hormone (LH) [59]. FSH is responsible for the development of ovarian follicles, one of which becomes the dominant follicle and releases an egg during the ovulation phase. The ovulatory follicle secretes estradiol [60]. FSH is also responsible for the activity of aromatase in granulosa cells involved in the synthesis of estrogens, as well as for the formation of LH receptors on the membrane of these cells and in the corpus luteum [61]. The maximum surge of LH causes ovulation, in the place of which, after ovulation, the corpus luteum is formed, which produces progesterone [62].

Lifestyle and various environmental factors can impair female fertility and, consequently, may affect the functions of their reproductive system [63]. Light physical exercise usually does not cause changes in the regularity of cycles and does not lead to anovulation, but it can lead to a decrease in the level of ovarian steroid hormones. Weight loss due to dietary restrictions can also have this effect [64]. In women who perform intense physical activity, it is observed that the pulsatile secretion of GnRH is irregular, and the concentration of LH is reduced [65]. An important factor affecting fertility is stress. However, studies on the effects of acute and short-term stress on estrogen and progesterone secretion are inconclusive. Studies conducted on monkeys, rodents, and sheep, among others, have shown that this type of stress causes both an increase and decrease in the production of these hormones, respectively [66,67]. Besides, environmental factors are affecting the concentration of estradiol and progesterone. In premenopausal women, for instance, switching from a high-fat to a low-fat diet leads to a decrease in estradiol levels [68]. There was no relationship between fiber intake and the concentration of sex steroid hormones [69]. The relationship between progesterone levels and alcohol consumption still remains unexplained [70]. Studies in which the presence of miRNA in granular and cumulus cells were detected, underlined the communication between these cells of ovarian follicles [71]. The interaction of miRNA and mRNA in the network of signaling pathways have been shown to affect the regulation of ovulation rates in goats. These signaling pathways, which were involved in fertility traits of goats, were found based on the miRNA profiles and transcriptome in the ovaries [72]. MiRNAs are involved in the regulation of mammalian reproduction, regulation of follicular development, and affect the synthesis and release of steroid hormones from the ovaries, for example, estradiol. MiRNAs are also of importance in the diagnosis of ovarian cancer and prognosis of ovarian chemotherapy. Identification of miRNAs helps to define reproductive problems [73]. Another interesting and crucial cornerstone for the interactions between the oocyte and the surrounding cumulus cells is the cell-to-cell communication. This special communication allows a fast signal exchange between the oocyte and the surrounding follicular cells and is possible due to several connexins and pannexins. The detailed role of these two latter mentioned proteins has been reviewed recently elsewhere [74].

## 4. Implementations of Stem Cells for Reproductive and Regenerative Medicine

Ovarian stem cells and oocyte precursor cells in ovaries enabled novel therapeutic strategies, such as the autologous germline mitochondrial energy transfer (AUGMENT) for oocytes of advanced age [75]. Other assisted reproductive technologies are in vitro fertilization (IVF), intracytoplasmic sperm injection (ICSI), mitochondrial transfer, or total cytoplasm transfer, to replace the aged or pathologic cytoplasm of an oocyte [75]. When discussing the transcriptional mechanisms that enable the transition from a silenced oocyte to a transcriptionally active embryo, it is worth mentioning stem cells, which are of increasing interest in science. Transcription studies during ZGA have shown that the genes activated during this process overlap with the genes that lead to the generation of iPSCs (induced pluripotent stem cells), a pluripotent cell obtained by the reprogramming of gene expression of differentiated cells [48]. It has also been reported that these genome activators are also pluripotency regulators [5,50]. An example of this is the mammalian OCT4 mentioned above, or SOX2 [48,76]. Moreover, tests of these factors on mouse fibroblasts have shown that their overexpression can generate iPS-like cells or ESCs (embryonic stem cells), which has also been demonstrated in human cells [77]. It has been postulated that OCT4 appears to be “the strongest” factor in promoting iPSC programming [50]. In the ovaries of some mammals, e.g., in humans and rodents, the presence of ovarian stem cell-like cells (OSCs) have been described that seem to generate new functional oocytes capable of producing fertile offspring after in vitro procedures (Figure 3) [78,79]. The differentiation of OSCs into different stages of oogenesis can be followed up by labeling the OCT4 factor or the meiotic marker SCP3 with GFP [78]. However, in addition to the data on humans and rodents that are taken into consideration for research on OSCs, data on bovine oocytes have also been provided in which the pluripotency markers OCT4 and SOX2 have been detected [78,80]. Moreover, human ESCs can differentiate into OSCs and also into granulosa cells which, as mentioned above, are crucial for proper oocyte development. Researchers indicate that the origin of OSCs and its derivatives may occur from various sources, i.e., bone marrow, ovarian cortex, or ovarian epithelium. On the other hand, ovarian stem cell-like cells obtained from marmoset monkeys were positive for germ cell markers and for the SCP3 meiotic marker, which is used to identify OSCs undergoing postnatal oogenesis. Depending on the type of material, various isolation methods are used, for instance, fluorescence-activated cell sorting (FACS), magnetic-activated cell sorting (MACS), immunocytochemistry, or anti-DDX4 antibody cell sorting, although most studies focus on FACS and MACS as isolation methods. Isolated ovarian stem cell-like cells from rabbits, monkeys, sheep, and menopausal women may form structures similar to oocytes or ESCs [78]. This may suggest that there is a chance for peri-menopausal or sub-fertile women to become pregnant [80]. This is especially important in the face of approximately 80 million women who struggle with the problem of infertility, and estimates indicate that this number will continue to grow due to changing lifestyles and environmental threats [80].

Notably, the technology of iPSCs started in 2006 when a report was made about cells similar to ESCs that can be generated from somatic cells using four transcription factors, OCT4, SOX2, KLF4, and MYC, which are so called Yamanaka factors. The first studies focused on mouse fibroblasts, but about a year later, success using human fibroblasts had been reported [81,82]. Generally speaking, iPSCs behave functionally the same as ESCs, though their origins are completely different. ESCs are obtained from the early embryonic stage, which is synonymous with the destruction of the embryo, and the time when they can be collected is limited to the 5–14th day of the blastocyst stage [83]. In the second case, we are talking about somatic cells that have been introduced at the pluripotency stage [77]. This has a direct impact on the amount of data we have on these two types of cells, due to ethical concerns in the use of ESCs. The possibility of inducing somatic cells into pluripotent stem cells opens the way to their wider application in research. In addition, the fact that the same factors are used, such as OCT4, increases the availability of stem cells, which are otherwise limited to specific tissues. In front of this information, it seems that the use of iPSCs may constitute a serious branch in developing xenotransplantation. Researchers are already trying to make cross-species cell transplants in rodents, which has proven to be effective and safe for donors [84]. Just a few years ago, researchers conducted studies on chimeric islets that were isolated from the pancreas of rats, which were complemented with murine iPSCs, and then transplanted into diabetic mice. Not only did this procedure show short-term success, the grafts also survived for over a year. What is more relevant for humans is to use this technology to generate interspecies organs for xenotransplantation [85].

It turns out, however, that despite promising theoretical assumptions, xenotransplantation, especially of whole organs, is an extremely complex and complicated task, which is not always successful. Due to ethical controversies, approval of such a trans-plantation was obtained on the basis of the belief that an experimental transplant would not be worse than continuing traditional treatment. It has been reported that a porcine heart with a modification of 10 genes has been designed for xenotransplantation. Among the latter-mentioned genetic modifications, a galactose-α-1,3-galactose knock-out (GTKO) was induced, a knock-out of the alpha1,3-galactosyltransferase (GT) enzyme leading to the deletion of the immunogenic galactose-α-1,3-galactose (Gal), which is supposed to have an anti-immunogenic activity or the endothelial cell protein C receptor (EPCR) which, by activating protein C, is supposed to have an anticoagulant effect [86]. As a result of the xenotransplantation, the heart obtained from a pig was able to sustain the patient’s life for 7 weeks. Doctors during this time conducted therapy aimed at depletion of T and B cells and therapy based on CD40, which prevented rejection of the graft modified in this way, which was later confirmed by endomyocardial biopsies. On day 49 after transplantation, acrocyanosis developed, suggesting decreased cardiac output, and the patient’s examination showed a low level of mixed venous oxygen saturation—33% [87,88]. A later autopsy showed that the weight of the heart had almost doubled since transplantation. Many complications that occurred after this event led scientists to the conclusion that there was irreversible damage to the xeno-heart, as a result of which, together with the patient’s family, a decision was made on the 60th day after the transplantation to discontinue further support [87].

Despite the improving situation in the availability of organs for transplantation in re-cent years, many patients will die while waiting for life-saving surgery due to the unavailability of organs. To be able to perform a transplant, the donor’s organ must be “matched” to the recipient according to the blood group or the major histocompatibility system (MHC) [89]. Therefore, in the future, using the knowledge gained from the study of ZGA and iPSC, protocols for 3D cell cultures, we may be able to “print” tissue fragments or entire organs using iPSCs obtained from other mammals. It is noteworthy that the implementation of stem cells as a potential therapeutic strategy is multifaceted, especially in regenerative medicine. For instance, they could be beneficial for patients as an in vivo approach using tissue engineering methods, which are considered as more advantageous therapeutic methods than the use of somatic cells for such a procedure, and they have greater ability to proliferate or later succumb to the aging [90]. Multipotent hematopoietic stem cell (HSC) transplantation deserves special attention as the most popular cell therapy. It consists of the transplantation of cells generated from bone marrow, peripheral blood, or umbilical cord blood for the treatment of diseases such as leukemia or anemia. Scientists are still working on cell therapies that will replace arthroplasty and pathologies associated with this disease affecting high-performance athletes and the elderly. The wide potential in the use of stem cells can be seen in diseases that, according to the current state of knowledge, are considered incurable, i.e., Alzheimer’s disease (AD) or Parkinson’s disease (PD). Previous studies postulate that by 2050, approximately 30% of the population will suffer from age-related diseases such as AD or PD [91]. Interestingly, with stem cell therapy it appears possible to delay the progression of these diseases [92]. Though on first glance, the use of stem cells seems to be an ideal solution for numerous pathologies, there are limitations too. For instance, bone marrow-derived mesenchymal stem cells (BMMSCs) have a limited ability to self-renewal and the procedure of obtaining them is invasive. They also apply to iPSCs, since due to the high rate of apoptosis, there is a risk that they will form tumors and their limited ability to repair DNA may lead to mutations [93,94].

## 5. Conclusions and Outlook

In conclusion, there is no doubt that more studies are needed to elucidate the regulatory pathways in late oogenesis in mammalian species. It would be further interesting to elucidate which genes are differentially expressed in oocytes generated from women of different maternal ages. It is noteworthy that epigenetic changes do also play a role in early embryonic development and oogenesis. Especially, heterochromatin in oocytes and the DNA methylation landscape experience changes, and it is well-described that the oocyte undergoes re- and de-methylation. More precisely, there are two waves of genome-wide reprogramming, firstly in germ cells and secondly in preimplantation embryos. Epigenetic reprogramming in oocytes appears to be crucial for imprinting. All in all, due to ethical restrictions and due the controversial discussion about ovarian stem cells and factors that are influencing the oocyte developmental competence or influencing early embryonic development in humans, more data are required to shed light on pathways that are involved in oogenesis and embryogenesis. It is an interesting fact that OSCs appear to be present adult ovaries and, as a consequence, may suggest that the somatic cells are comprising the niche for the OSC change with maternal age, which leads to menopause in women. There is no doubt that the cumulus and granulosa cells provide a crucial ovarian microenvironment during oogenesis and their functioning is affected by ageing. Previous research by the Tilly group provided strong evidence that OSCs generated from aged ovaries are capable to give rise to functional oocytes once they are transplanted into an ovary of younger counterparts. Therefore, future research directions could focus on neo-oogenesis and primordial follicle assembly with regard to maternal aging, and maybe someday, OSCs will be a therapeutic target for sub- and infertile women, or to delay menopause, or to restore the ovarian microenvironment in patients after cancer therapy.

## Figures and Tables

**Figure 1 ijms-24-06837-f001:**
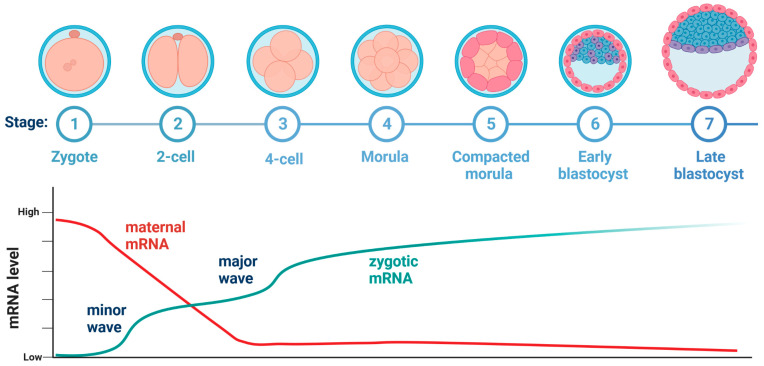
Levels of maternal and zygotic mRNA with regard to embryonic stages in the murine species.

**Figure 2 ijms-24-06837-f002:**
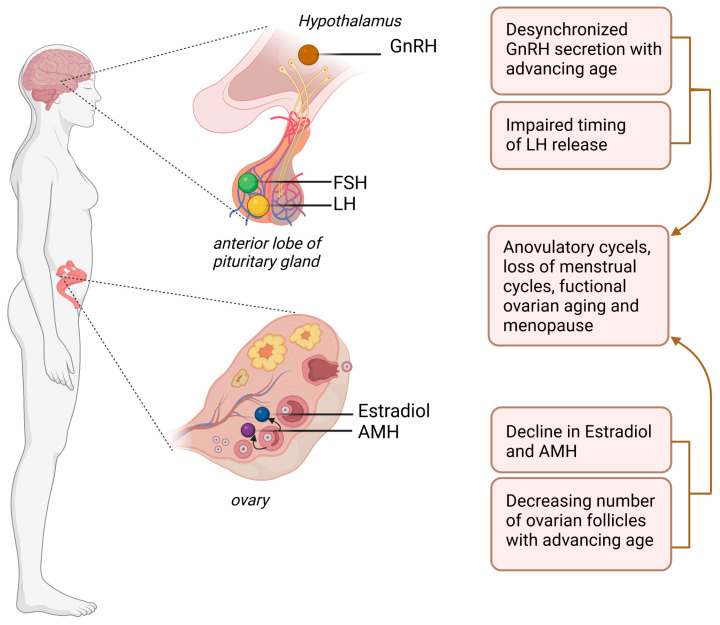
Scheme showing functional ovarian aging and the hypothalamic, hypophyseal, and ovarian hormones.

**Figure 3 ijms-24-06837-f003:**
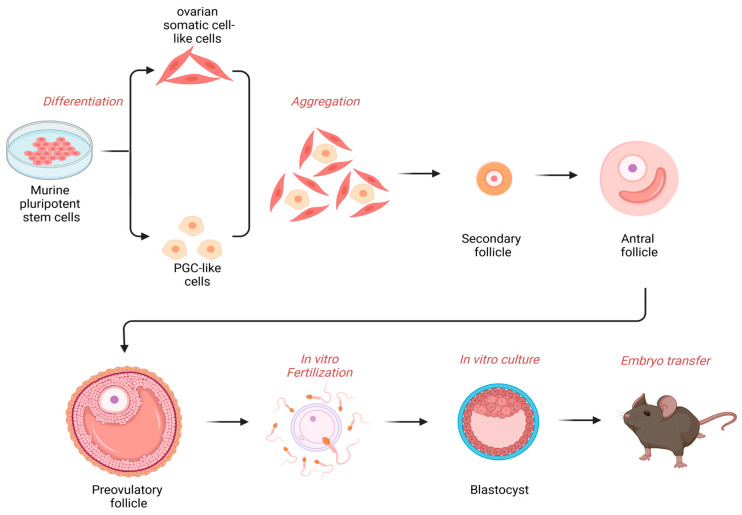
Scheme showing the in vitro production of functional oocytes from pluripotent stem cells in the murine animal model.

## Data Availability

Not applicable.

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
