# Peer review of "Oogenesis in Women: From Molecular Regulatory Pathways and Maternal Age to Stem Cells"

_ijms, 2023, doi:10.3390/ijms24076837_

Round 1

Reviewer 1 Report

The title of the manuscript is good but the authors should explain about the novility of their work (The importance and the novility of their work in under question). English language is simple and acceptable. Figures are good in quality.The multiple and middle-sentence references and sentences without proper reference are the citation problems in this manuscript.

1. The title of the manuscript is "Molecular regulatory pathways during oogenesis in women: How to deal with increasing maternal age? "

The first part of the title have used frsquently before (Molecular regulatory pathways during oogenesis in women). The second part is (How to deal with increasing maternal age?), but in the main text, you have focused on stem cells and their potential to counteract aging-related processes. Thus, the title can not represent the idea of main text 

completely. Please reconsider the title. If you think opposite, please explain why?

2. Would you please tell what is the novility of the manuscript?

Please explain what is the difference of your work with the manuscript entitled " Molecular control of oogenesis

Flor Sánchez, Johan Smitz

Biochimica et Biophysica Acta (BBA)-Molecular Basis of Disease 1822 (12), 1896-1912, 2012 "

3. All over the manuscript, there are some multipple and middle-sentence references

Please reconsider all of them

4. Page 6, line 244

Please tell that this title is a question? Or is a proved sentence? It seems a little vague

5. Why you have not written about the effects of aging process on oogenesis and ovulation in a separate part?

6. Why you have not mentioned about "The animal studies that are related to the title of your work" in a separate part? (Human and animal studies should be discussed separately)

7. Why you have not spoken about the future insight of using stem cells in related with the title of the manuscript in a separate part?

8. What about the limitations of stem cells in related to the title? Why you have not discussed about it in a separate part?

9. Please check and adjust the "Reference list" based on the regulations of reference list of journal. (Titles, doi, the name of journal and ... )

Author Response

MS ID: ijms-2290434

Dear Academic Editor,

Dear Reviewers,

Thank you for inviting us to respond to the very thoughtful and constructive reviewer comments. We greatly appreciate the Reviewers time and believe our revised manuscript has become more well-rounded as a result.

We have incorporated all suggestions throughout the manuscript which are high-lighted in red in the manuscript. Below is a point-by-point response to reviewers’ comments to clarify which edits were made.

We are happy to respond to additional requests if they arise.

Sincerely,

Mariusz Skowronski

Pawel Kordowitzki

Please note our following explanations:

Detailed answers to Reviewer 1

REVIEWER: "1. The title of the manuscript is "Molecular regulatory pathways during oogenesis in women: How to deal with increasing maternal age? "

The first part of the title have used frsquently before (Molecular regulatory pathways during oogenesis in women). The second part is (How to deal with increasing maternal age?), but in the main text, you have focused on stem cells and their potential to counteract aging-related processes. Thus, the title can not represent the idea of main text completely. Please reconsider the title. If you think opposite, please explain why?"

ANSWER: We want to thank the Reviewer for his/her thorough review of our manuscript. In response, we acknowledge these important points, and we have changed the title and added the following sentences (L18-L27,L255-302, L318-323):

Title:  Oogenesis in women: From molecular regulatory pathways and maternal age to stem cells

L18-L27:

More precisely, the longer a given oocyte is in the waiting-line to be ovulated from menarche to menopause, the longer is the duration from oogenesis to fertilization, and therefore, the lower are the chances of success to form a viable embryo. The age of menarche in girls ranges from 10 to 16 years, and the age of menopause in women ranges from approximately 45 to 55 years. Researchers are paying attention, on the regulatory pathways which are impacting the oocyte at the very beginning during oogenesis in fetal life, to discover genes and proteins which could be crucial for the oocyte’s life-span. Due to the general trend in industrialized countries in the last three decades, women are giving birth of their first child in their thirties. Therefore, maternal age has become an important factor impacting oocytes developmental competence, since the higher woman’s age the higher are the chances of miscarriage due to several causes such as aneuploidy.

(L255-302):

Concentrations of steroid hormones - estradiol and progesterone determine the potential fertility of women and affect many issues related to their health [53].The dominant sex steroid hormones in women are estrogens, such as estrone (E1), estradiol (E2), and estriol (E3). Estradiol is the main circulating sex hormone before menopause, while estrone plays an important role after menopause [54]. The direct substrate for the synthesis of estrogens are testosterone and androstenedione [55]. Here, SNARE proteins (soluble NSF attachment receptor) are involved in the transport of cholesterol from lipid cells. Cholesterol is then converted to pregnenolone for further production of dehydroepiandrosterone (DHEA), which is an androgen precursor [56]. The site of estrogen synthesis in non-pregnant women mainly the ovary, while in pregnant women it is the placenta [54]. Estrogens do among others play a crucial role during the regulation of the menstrual cycle [57]. In addition, these hormones affect calcium and lipid metabolism and affect protein synthesis [58]. The level of circulating estradiol and the Anti-Müllerian-Hormone (AMH) is affected by the reproductive status of women. Especially the AMH level decreases with advancing age and with an advancing decrease of remaining follicles, mirroring therefore the ovarian reserve (Fig.2) [54]. When discussing the hormonal regulation of the female reproductive cycle, the hypothalamic-pituitary-ovarian axis should also be mentioned. An important hormone of this axis is Gonadotropin Releasing Hormone (GnRH), which is a hypothalamic hormone stimulating the anterior lobe of the pituitary gland to release follicle-stimulating hormone (FSH) and luteinizing hormone (LH) [59]. FSH is responsible for the development of ovarian follicles, one of which becomes the dominant follicle and releases an egg during the ovulation phase. The ovulatory follicle secretes estradiol [60]. FSH is also responsible for the activity of aromatase in granulosa cells involved in the synthesis of estrogens, as well as for the formation of LH receptors on the membrane of these cells and in the corpus luteum [61]. The maximum surge of LH causes ovulation, in the place of which, after ovulation, the corpus luteum is formed, which produces progesterone [62].Lifestyle and various environmental factors can impair female fertility and, consequently, may affect the functions of their reproductive system [63]. Light physical exercise usually does not cause changes in the regularity of cycles and does not lead to anovulation, but it can lead to a decrease in the level of ovarian steroid hormones. Weight loss due to dietary restrictions can also have this effect [64]. In women who perform intense physical activity, it is observed that the pulsatile secretion of GnRH is irregular and the concentration of LH is reduced [65]. An important factor affecting fertility is stress. However, studies on the  effects of acute and short-term stress on estrogen and progesterone secretion are inconclu-sive. Studies conducted on monkeys, rodents, and sheep, among others, have shown that this type of stress causes both an increase and decrease in the production of these hor-mones, respectively [66,67]. Besides, environmental factors are affecting the concentration of estradiol and progesterone. In premenopausal women, for instance switching from a high-fat to a low-fat diet leads to a decrease in estradiol levels [69]. There was no relation-ship between fiber intake and the concentration of sex steroid hormones [70]. The relation-ship between progesterone levels and alcohol consumption still remains unexplained [71].

L318-323:

Ovarian stem cells and oocyte precursor cells in ovaries enabled novel therapeutic strategies, such as the autologous germline mitochondrial energy transfer (AUGMENT) for oocytes of advanced age [76]. Other assisted reproductive technologies are the in vitro fertilization (IVF), the intracytoplasmic sperm injection (ICSI), mitochondrial transfer or the total cytoplasm transfer, to replace aged or pathologic cytoplasm of an oocyte [76].

REVIEWER: "2. Would you please tell what is the novility of the manuscript?

Please explain what is the difference of your work with the manuscript entitled " Molecular control of oogenesis, Flor Sánchez, Johan Smitz, Biochimica et Biophysica Acta (BBA)-Molecular Basis of Disease 1822 (12), 1896-1912, 2012 "

3. All over the manuscript, there are some multipple and middle-sentence references. Please reconsider all of them."

ANSWER: We want to thank the Reviewer for this comment. In response, we have carefully updated the references.  With regard to  point 2, we would like to explain that the above mentioned reference comprehensively deals with oogenesis by pointing to various transcriptional components without describing what happens after the oocyte is released from the ovarian follicle. Our text focuses strictly on the molecular mechanisms that we recognize as crucial when studying the process of late oogenesis and early embryogenesis, which can be related to the research and use of stem cells and the potential use of this knowledge in the treatment of not only the problem of infertility. Therefore, the novelty of our Review is that we tried to extrapolate the current knowledge on oogenesis to the topic of stem cells. Furthermore, our review paper deals strictly with the molecular mechanisms involved in the process of late oogenesis and early embryogenesis and the information we have included in it has so far been analyzed separately. This makes our work a source with a wide range of knowledge about the transcription mechanisms involved in the processes we describe. In addition, our noticing the similarities in gene activation during ZGA and iPSC formation, allowed us to show a new way of researching on stem cells that is widely used by scientists around the world, and demonstrating a new approach expands the current state of knowledge and strengthens the innovative nature of our work.

REVIEWER: "4. Page 6, line 244: Please tell that this title is a question? Or is a proved sentence? It seems a little vague."

ANSWER: We thank the Reviewer for his/her comment, which we appreciate.  In response, we have changed the title into (L318): 

  1. Implementations of stem cells for reproductive and regenerative medicine

REVIEWER: "Why you have not written about the effects of aging process on oogenesis and ovulation in a separate part?

ANSWER: We thank the Reviewer for his/her question. We would kindly like to clarify that the aim of our work was to gather knowledge about transcriptional activity during late oogenesis and early oogenesis in order to understand the basics and characteristics of the changes taking place then. What's more, we intended to provide novel therapeutic strategies by pointing out a group of cells able to provide theraoeutic strategies for women's infertility or renew the ovarian microenvironment. In our opinion, it was not necessary to describe the factors affecting the process in a seperate part, as our target is to deal with the consequences of ovary malfunction not prevent it, although it is definitely an interesting area. Describing only the influence of old age would not be enough to exhaust the topic, which requires much more attention and deserves a separate paper. Nevertheless, we have added new references, a new figure 2, and more information on fertilitly, aging and menopause in women. Please note the changes in the attached manuscript.

REVIEWER: "6. Why you have not mentioned about "The animal studies that are related to the title of your work" in a separate part? (Human and animal studies should be discussed separately).

7. Why you have not spoken about the future insight of using stem cells in related with the title of the manuscript in a separate part?

8. What about the limitations of stem cells in related to the title? Why you have not discussed about it in a separate part?"

ANSWER: We thank the Reviewer for his/her interesting and stimulating questions. Based on the data we collected, we can conclude that the action of the factors described by us, such as CPEB responsible for the polyadenylation mechanism in human and mouse oocytes, or the ribosome binding protein RBP, which determine almost all RNA life (and have been described in detail for many model organisms) is the same for many species. Therefore, we decided to discuss them not separately for humans and animals, but as a general set of transcription processes that take place during oogenesis and embryogenesis. This was in geenral our porpuse not to create to many sections but discuss several topics in one section. In response to question 7 and 8, we decided to add more information, as follows (L414-434):

Noteworthy, the implementation of stem cells as a potential therapeutic strategy is multifaceted, especially in regenerative medicine. For instance, they could be beneficial for patients as an in vivo approach using tissue engineering methods, which are considered as more advantageous therapeutic methods than the use of somatic cells for such a procedure, and they have greater ability to proliferate or later succumb to the aging [91]. Multipotent hematopoietic stem cell (HSC) transplantation deserves special attention as the most popular cell therapy. It consists of the transplantation of cells generated from bone marrow, peripheral blood, or umbilical cord blood for the treatment of diseases such as leukemia or anemia. Scientists are still working on cell therapies that will replace arthroplasty and pathologies associated with this disease affecting high-performance athletes and the elderly. The wide potential in the use of stem cells can be seen in diseases that, according to the current state of knowledge, are considered incurable, i.e. Alzheimer's disease (AD) or Parkinson's disease (PD). Previous studies postulate that by 2050, approximately 30% of the population will suffer from age-related diseases such as AD or PD [92]. Interestingly, with stem cell therapy it appears possible to delay the progression of these diseases [93]. Though on first glance, the use of stem cells seems to be an ideal solution for numerous pathologies, there are limitations, too. For instance, bone marrow-derived mesenchymal stem cells (BMMSCs) have a limited ability to self-renewal and the procedure of obtaining them is invasive. They also apply to iPSCs, since the high rate of apoptosis, there is a risk that they will form tumors and their limited ability to repair DNA may lead to mutations [94,95].

REVIEWER: "9. Please check and adjust the "Reference list" based on the regulations of reference list of journal. (Titles, doi, the name of journal and ... )"

ANSWER: Thank you for this comment which has been addressed accordingly.

Reviewer 2 Report

The manuscript deal with an important problem, however, I have the impression that the topic is more expansive and needs to be additionally completed according suggestions

Basic information is needed to introduce readers to the problem

Give information on:

- the mean age of first menarche in girls

- the mean age of menopause

-age of the women planning their first child

- ART procedures that are utilized for the aging woman; give woman age and information about its successful implementation

- diseases related to maternal age

-sex steroid hormone regulation and precise action

-factors especially of chemical properties present in the environment

Other information

What about:

 -miRNA, microvesicles  etc

-nongenomic regulation, fast signaling pathways, membrane sex steroid receptors

-data from monkeys, rodents other animal models

Author Response

MS ID: ijms-2290434

Dear Academic Editor,

Dear Reviewers,

Thank you for inviting us to respond to the very thoughtful and constructive reviewer comments. We greatly appreciate the Reviewers time and believe our revised manuscript has become more well-rounded as a result.

We have incorporated all suggestions throughout the manuscript which are high-lighted in red in the manuscript. Below is a point-by-point response to reviewers’ comments to clarify which edits were made.

We are happy to respond to additional requests if they arise.

Sincerely,

Mariusz Skowronski

Pawel Kordowitzki

Please note our following explanations:

Detailed answers to Reviewer 2

REVIEWER: "Give information on:

- the mean age of first menarche in girls

- the mean age of menopause

-age of the women planning their first child

-Diseases related to maternal age

ANSWER: We want to thank the Reviewer for his/her thorough review of our manuscript. In response, we acknowledge these important points, and we have added the following sentences (L18-L27):

More precisely, the longer a given oocyte is in the waiting-line to be ovulated from menarche to menopause, the longer is the duration from oogenesis to fertilization, and therefore, the lower are the chances of success to form a viable embryo. The age of menarche in girls ranges from 10 to 16 years, and the age of menopause in women ranges from approximately 45 to 55 years. Researchers are paying attention, on the regulatory pathways which are impacting the oocyte at the very beginning during oogenesis in fetal life, to discover genes and proteins which could be crucial for the oocyte’s life-span. Due to the general trend in industrialized countries in the last three decades, women are giving birth of their first child in their thirties. Therefore, maternal age has become an important factor impacting oocytes developmental competence, since the higher woman’s age the higher are the chances of miscarriage due to several causes such as aneuploidy. 

REVIEWER: "Give information on:

-ART procedures that are utilized for the aging woman; give woman age and information about its successful implementation."

ANSWER: We want to thank the Reviewer for this comment. In response, we decided adding only the common procedures and we have added the following sentences (L318-323):

Ovarian stem cells and oocyte precursor cells in ovaries enabled novel therapeutic strategies, such as the autologous germline mitochondrial energy transfer (AUGMENT) for oocytes of advanced age [76]. Other assisted reproductive technologies are the in vitro fertilization (IVF), the intracytoplasmic sperm injection (ICSI), mitochondrial transfer or the total cytoplasm transfer, to replace aged or pathologic cytoplasm of an oocyte [76]. 

REVIEWER: "Give information on:

-sex steroid hormone regulation and precise action

-factors especially of chemical properties present in the environment"

ANSWER: We thank the Reviewer for his/her comment, which we appreciate.  In response, we decided adding a new Figure 2 and the the following sentences (L255-302):

Concentrations of steroid hormones - estradiol and progesterone determine the potential fertility of women and affect many issues related to their health [53].The dominant sex steroid hormones in women are estrogens, such as estrone (E1), estradiol (E2), and estriol (E3). Estradiol is the main circulating sex hormone before menopause, while estrone plays an important role after menopause [54]. The direct substrate for the synthesis of estrogens are testosterone and androstenedione [55]. Here, SNARE proteins (soluble NSF attachment receptor) are involved in the transport of cholesterol from lipid cells. Cholesterol is then converted to pregnenolone for further production of dehydroepiandrosterone (DHEA), which is an androgen precursor [56]. The site of estrogen synthesis in non-pregnant women mainly the ovary, while in pregnant women it is the placenta [54]. Estrogens do among others play a crucial role during the regulation of the menstrual cycle [57]. In addition, these hormones affect calcium and lipid metabolism and affect protein synthesis [58]. The level of circulating estradiol and the Anti-Müllerian-Hormone (AMH) is affected by the reproductive status of women. Especially the AMH level decreases with advancing age and with an advancing decrease of remaining follicles, mirroring therefore the ovarian reserve (Fig.2) [54]. When discussing the hormonal regulation of the female reproductive cycle, the hypothalamic-pituitary-ovarian axis should also be mentioned. An important hormone of this axis is Gonadotropin Releasing Hormone (GnRH), which is a hypothalamic hormone stimulating the anterior lobe of the pituitary gland to release follicle-stimulating hormone (FSH) and luteinizing hormone (LH) [59]. FSH is responsible for the development of ovarian follicles, one of which becomes the dominant follicle and releases an egg during the ovulation phase. The ovulatory follicle secretes estradiol [60]. FSH is also responsible for the activity of aromatase in granulosa cells involved in the synthesis of estrogens, as well as for the formation of LH receptors on the membrane of these cells and in the corpus luteum [61]. The maximum surge of LH causes ovulation, in the place of which, after ovulation, the corpus luteum is formed, which produces progesterone [62].Lifestyle and various environmental factors can impair female fertility and, consequently, may affect the functions of their reproductive system [63]. Light physical exercise usually does not cause changes in the regularity of cycles and does not lead to anovulation, but it can lead to a decrease in the level of ovarian steroid hormones. Weight loss due to dietary restrictions can also have this effect [64]. In women who perform intense physical activity, it is observed that the pulsatile secretion of GnRH is irregular and the concentration of LH is reduced [65]. An important factor affecting fertility is stress. However, studies on the  effects of acute and short-term stress on estrogen and progesterone secretion are inconclu-sive. Studies conducted on monkeys, rodents, and sheep, among others, have shown that this type of stress causes both an increase and decrease in the production of these hor-mones, respectively [66,67]. Besides, environmental factors are affecting the concentration of estradiol and progesterone. In premenopausal women, for instance switching from a high-fat to a low-fat diet leads to a decrease in estradiol levels [69]. There was no relation-ship between fiber intake and the concentration of sex steroid hormones [70]. The relation-ship between progesterone levels and alcohol consumption still remains unexplained [71].

REVIEWER: "Give information on:

-miRNA, microvesicles etc

-nongenomic regulation, fast signaling pathways, membrane sex steroid receptors

-data from monkeys, rodents other animal models

ANSWER: We thank the Reviewer for his/her comments, which we appreciate.  In response, we decided adding the following sentences (L303-317 and L440-445):

 Studies in which the presence of miRNA in granular and cumulus cells were detected, underlined the communication between these cells of ovarian follicles [72]. The interaction of miRNA and mRNA in the network of signaling pathways have been shown to affect the regulation of ovulation rates in goats. These, signaling pathways which were involved in fertility traits of goats were found based on miRNA profiles and transcriptome in the ovaries of miRNAs [74]. MiRNAs are involved in the regulation of mammalian reproduction and regulation of follicular development and affects the synthesis and release of steroid hormones from the ovaries, for example estradiol. MiRNAs are also of importance in the diagnosis and prognosis of ovarian chemotherapy. Identification of miRNA helps to define reproductive problems [74]. Another interesting and crucial cornerstone for the interactions between the oocyte and the surrounding cumulus cells is the cell-to-cell communication. This special communication allows a fast signal exchange between the oocyte and the surrounding follicular cells and is possible due to several Connexins and Pannexins. The detailed role of these two latter mentioned proteins has been reviewed recently elsewhere [75].

Noteworthy, epigenetic changes do also play a role for early embryonic development and oogenesis. Especially, heterochromatin in oocytes and the DNA methylation landscape experience changes, and it is well-described that the oocyte undergoes re- and de-methylation. More precisely, there are two waves of genome-wide reprogramming, firstly in germ cells and secondly in preimplantation embryos. Epigenetic reprogramming in oocytes appears to be crucial for imprinting.

Round 2

Reviewer 1 Report

I do not have more suggestion.

Reviewer 2 Report

I accept corrections